# Repeated Social Defeat Exaggerates Fibrin-Rich Clot Formation by Enhancing Neutrophil Extracellular Trap Formation via Platelet–Neutrophil Interactions

**DOI:** 10.3390/cells10123344

**Published:** 2021-11-28

**Authors:** Takeshi Sugimoto, Hiroyuki Yamada, Naotoshi Wada, Shinichiro Motoyama, Makoto Saburi, Hiroshi Kubota, Daisuke Miyawaki, Noriyuki Wakana, Daisuke Kami, Takehiro Ogata, Masakazu Ibi, Satoaki Matoba

**Affiliations:** 1Department of Cardiovascular Medicine, Graduate School of Medical Science, Kyoto Prefectural University of Medicine, Kyoto 602-8566, Japan; sugimoto@koto.kpu-m.ac.jp (T.S.); wada-n@koto.kpu-m.ac.jp (N.W.); motoyama@koto.kpu-m.ac.jp (S.M.); msaburi@koto.kpu-m.ac.jp (M.S.); kbt-h@koto.kpu-m.ac.jp (H.K.); torisan@koto.kpu-m.ac.jp (D.M.); nw0920@koto.kpu-m.ac.jp (N.W.); matoba@koto.kpu-m.ac.jp (S.M.); 2Department of Regenerative Medicine, Graduate School of Medical Science, Kyoto Prefectural University of Medicine, Kyoto 602-8566, Japan; dkami@koto.kpu-m.ac.jp; 3Department of Pathology and Cell Regulation, Graduate School of Medical Science, Kyoto Prefectural University of Medicine, Kyoto 602-8566, Japan; ogatat@koto.kpu-m.ac.jp; 4Department of Pharmacy, Kinjo Gakuin University, Nagoya 463-8521, Japan; ibi@kinjo-u.ac.jp

**Keywords:** depression, cardiovascular disease, thrombosis, neutrophil extracellular trap, platelet, CD11b/CD18 integrin

## Abstract

Depression is an independent risk factor for cardiovascular disease (CVD). We have previously shown that repeated social defeat (RSD) exaggerates atherosclerosis development by enhancing neutrophil extracellular trap (NET) formation. In this study, we investigated the impact of RSD on arterial thrombosis. Eight-week-old male wild-type mice (C57BL/6J) were exposed to RSD by housing with larger CD-1 mice in a shared home cage. They were subjected to vigorous physical contact daily for 10 consecutive days. After confirming depression-like behaviors, mice underwent FeCl_3_-induced carotid arterial injury and were analyzed after 3 h. Although the volume of thrombi was comparable between the two groups, fibrin(ogen)-positive areas were significantly increased in defeated mice, in which Ly-6G-positive cells were appreciably co-localized with Cit-H3-positive staining. Treatment with DNase I completely diminished exaggerated fibrin-rich clot formation in defeated mice. Flow cytometric analysis showed that neutrophil CD11b expression before FeCl_3_ application was significantly higher in defeated mice than in control mice. In vitro NET formation induced by activated platelets was significantly augmented in defeated mice, which was substantially inhibited by anti-CD11b antibody treatment. Our findings demonstrate that RSD enhances fibrin-rich clot formation after arterial injury by enhancing NET formation, suggesting that NET can be a new therapeutic target in depression-related CVD.

## 1. Introduction

Depression is a leading cause of disability worldwide and a major contributor to the global burden of disease [1]. Meta-analyses have shown that depression is associated with a high risk of coronary heart disease and increased mortality after myocardial infarction [2,3,4,5]. Integration of various factors is associated with atherosclerotic cardiovascular disease (CVD). However, the precise mechanisms of depression-related CVD development are unclear [6,7]. Previous studies have reported that socially defeated mice exhibit increased peripheral inflammatory monocytes and granulocytes [8,9]. In a clinical study, the neutrophil–lymphocyte ratio was positively associated with the severity of depression [10]. Neutrophils are a heterogeneous cell population that communicate with other immune and non-immune cells during acute and chronic inflammation [11]. These findings suggest that neutrophils are associated with an increased risk of CVD in patients with depression.

Recent studies have shown that neutrophils are closely implicated in atherosclerosis through induction of neutrophil extracellular traps (NETs), a new type of neutrophil cell death [12,13,14]. Warnatsch et al. have shown that cholesterol-crystal-induced NET formation promotes the transcription of immature interleukin (IL)-1β in macrophages and that inhibition of NET formation by DNase I treatment significantly reduces plaque size as compared to that in apoE-deficient (apoE^−/−^) controls [15]. Liu et al. showed that myeloid-lineage-specific deletion of peptidyl arginine deiminase 4, a nuclear enzyme for histone citrullination, significantly reduced atherosclerosis development and diminished NET formation [16]. Furthermore, Silvestre-Roig et al. revealed that activated vascular smooth muscle cells (VSMCs) promote the release of histone H4 by neutrophils via NET formation, which exerts cytotoxic effects on VSMCs, leading to increased plaque vulnerability [17]. Based on these findings, we hypothesized that NET formation plays a more critical role in depression-related CVD than in conventional risk-factor-related CVD.

Tsankova et al. have shown that exposure to repeated social defeat (RSD) leads to the development of depressive-like behaviors in animal models [18]. In our previous study, RSD evoked depressive-like behaviors in apoE^−/−^ mice and promoted atherosclerosis development accompanied by a significant increase in NET formation within the plaque, which was completely inhibited by treatment with DNase I [19]. In this study, we showed for the first time that RSD enhanced fibrin-rich clot formation after arterial injury in wild-type (WT) mice, which was completely inhibited by treatment with DNase I treatment. In vitro NET formation induced by activated platelets was significantly augmented in neutrophils of defeated mice compared to that in neutrophils of control mice, while platelet aggregation was comparable between the two groups. Along with the conceptual shift from ruptured plaques to eroded plaques, the clinical significance of NET for/mation has emerged as a therapeutic target for preventing acute coronary syndrome (ACS) [20,21,22]. Our findings suggest that NET formation plays a crucial role not only in the development of atherosclerosis but also in the onset of ACS in patients with depression. These findings provide novel mechanistic insights into the role of NET formation in depression-related CVD through platelet–neutrophil interactions.

## 2. Materials and Methods

### 2.1. Repeated Social Defeat Model

Male WT mice (C57BL/6J) and male CD-1 mice were purchased from Shimizu Laboratory Supplies Co., Ltd. (Kyoto, Japan), and maintained on a normal diet (12.0% fat, 28.9% protein, 59.1% carbohydrate; Oriental Yeast Co., Tokyo, Japan). Eight- to ten-week-old WT mice were exposed to RSD according to the protocol reported by Golden et al. [23] with modifications as described below. After screening aggressor CD-1 mice, each CD-1 resident mouse was housed with a WT intruder mouse. The two animals were separated by a perforated partition, which allowed for continuous visual, auditory, and olfactory contact with no physical interaction. They were repeatedly subjected to vigorous physical contact for 5–10 min every day for 10 consecutive days (Appendix A). During the 10 days of psychological stress, the intruder WT mouse was exposed daily to a novel resident’s home cage to prevent habituation to the resident aggressor. Control mice were housed in the same type of cage without physical contact with CD-1 resident mouse. Animals were housed in a room maintained at 22 °C under a 12 h light/dark cycle and provided with drinking water ad libitum.

### 2.2. Behavior Analysis

Behavioral tests were performed as previously described [24,25,26]. (1) Tail suspension test: Mice were suspended from their tails and the immobility time during a 6 min period was measured using a charge-coupled device (CCD) video camera (JIN-608AC; Kyohritsu Electronic Industry Co., Ltd., Osaka, Japan). (2) Social interaction test: To examine their social approach toward an unfamiliar mouse, the time spent in the interaction zone when the target was absent or present was measured using a CCD video camera. The social interaction ratio (SIR) was obtained by dividing the interaction time spent by the mice in the presence of the target by the time spent in the absence of the target. RSD-exposed mice exhibiting an SIR of <1.0 were considered as defeated mice with depression-like behaviors, while non-exposed mice exhibiting an SIR of >1.0 were considered as control mice.

### 2.3. FeCl_3_-Induced Clot Formation Model

After behavior analysis, 12-week-old mice were anesthetized with isoflurane (2%, 0.2 mL/min) using an anesthetic vaporization instrument (PITa-Quark; Sanko Manufacturing Co., Ltd., Saitama, Japan) connected to a nose cone throughout surgery. According to previously reported methods [27], the left common carotid artery was exposed by dissecting the surrounding fatty tissue. A filter paper (0.5 × 1.0 mm) saturated with 10% FeCl_3_ was applied on the adventitial surface of the carotid artery. After 3 min, the paper was removed, and the wound was washed with 0.9% sodium chloride (NaCl) three times before the skin incision was sutured. For DNase I treatment, mice were injected intravenously with 120 U DNase I (4536282001, Sigma-Aldrich St. Louis, MO, USA) in 100 μL of 0.9% NaCl solution, as previously described [15]. Control mice were only administered 100 μL of 0.9% NaCl. The effect of anesthesia was confirmed by the lack of tail pinch response, which was closely monitored throughout the procedure with a fine adjustment of isoflurane concentration to maintain an adequate depth of anesthesia. After recovery from anesthesia, monitoring was intensively continued for behavioral signs of postoperative pain, which was managed with ready-to-use lidocaine ointment.

### 2.4. Hemodynamic Analysis

Blood pressure and heart rate were measured under conscious and unrestrained conditions using a programmable sphygmomanometer (BP-98A; Softron, Tokyo, Japan). Unanesthetized mice were placed in a small holder mounted on a thermostatically controlled warming plate, which was maintained at 37 °C during measurement. Blood pressure and heart rate were measured three times while the mice were still in the holder.

### 2.5. Thrombi Measurement and Histological Analysis

Three hours after FeCl_3_ application, the left carotid artery was removed following transcardial perfusion with 0.9% NaCl and embedded in an optimal cutting temperature compound (OCT, Tissue-Tec, Funakoshi Co., Ltd., Tokyo, Japan) and quickly frozen in liquid nitrogen. The volume of thrombi was calculated by integrating the cross-sectional area of the thrombus-like lesion from 10 to 20 locations at 200 µm intervals. Hematoxylin and eosin staining was performed, and the slides were examined using light microscopy.

### 2.6. Immunohistochemical Analysis

Three sections (100 µm intervals) were prepared from the middle portion of the thrombus and immunohistochemically stained. For the immunological staining of fibrin(ogen), anti-fibrin(ogen) antibody (1:200, ab63983; Abcam, Cambridge, MA, USA) and Alexa Fluor 488–conjugated secondary antibody (Thermo Fisher Scientific, Waltham, MA, USA) were used. For Ly-6G, Alexa Fluor 647–conjugated anti-mouse Ly-6G (1:400, BD127626; BioLegend, San Diego, CA, USA) was used. For CD42b, anti-CD42b antibody (1:1000, ab104704; Abcam) and Alexa Fluor 488–conjugated secondary antibody (Thermo Fisher Scientific) were used. For Cit-H3, anti-Cit-H3 antibody (1:250, ab5103; Abcam, Cambridge, MA, USA) and Alexa Fluor 488–conjugated secondary antibody (Thermo Fisher Scientific) were used. Nuclei were labeled using 4’, 6-diamidino-2-phenylindole (excitation wavelength: 360 nm, fluorescence wavelength 461 nm, 62248; Thermo Fisher Scientific) or SYTOX Green (excitation wavelength: 504 nm, fluorescence wavelength 523 nm, 33025; Thermo Fisher Scientific). Sections were examined using an LSM 510 META confocal microscope (Carl Zeiss, Jena, Germany). Non-immune rabbit immunoglobulin G and polyclonal isotype control were used as negative controls of fibrin(ogen), CD42b, and Cit-H3. Positive staining was evaluated using ImageJ software v1.50i (https://imagej.nih.gov/ij/index.html, accessed on 20 October 2021).

### 2.7. Blood Leukocyte Counts

The blood was harvested from the left ventricle, collected in ethylenediaminetetraacetic acid or heparin tubes, and analyzed using the ADVIA 2120i hematology system (Siemens Healthcare, Erlangen, Germany).

### 2.8. Platelets Counts and In Vitro Aggregation

Platelet count analysis was performed using Thinka CB-1010 (ARKRAY, Inc., Kyoto, Japan). The platelet aggregation assay was outsourced to Nissei Bilis Co., Ltd., Shiga, Japan.

### 2.9. Preparation of Platelets and P-Selectin Expression Analysis

The blood was harvested from the left ventricle and collected in acid citrate dextrose (1 volume anticoagulant/9 volumes blood) tubes. To obtain an optimal amount of platelet-rich plasma (PRP), the blood was mixed with 1 volume of modified HEPES Tyrode’s buffer (140 mM NaCl, 2 mM KCl, 12 mM NaHCO_3_, 0.3 mM NaH_2_PO_4_, 1 mM MgCl_2_, 5.5 mM glucose, 5 mM HEPES; pH 6.7) containing 2 mM ethylene glycol tetraacetic acid and 0.35% HSA. After centrifugation at 150× *g* for 4 min, PRP was collected and pooled, and the remaining blood was centrifuged at 150× *g* for 4 min to collect PRP again. Then, PGI2 at a final concentration of 500 nM was added to pooled PRP, and platelets were pelleted by centrifugation at 1000× *g* for 4 min at 37 °C, as previously described [28]. P-selectin expression was analyzed by flow cytometry using PE-Cy7-conjugated CD41 (clone MWReg30; BD Biosciences, San Jose, CA, USA) and PE-conjugated CD62P (RMP-1; BD Biosciences) antibodies, under resting or thrombin (T7009; Sigma-Aldrich)-stimulated conditions (0.02 IU/mL, at 37 °C for 20 min), as previously described [29].

### 2.10. Flow Cytometry and Cell Sorting Analysis of Neutrophils

Bone marrow (BM) and peripheral blood (PB) neutrophils, except for mature BM and aged PB neutrophils, were analyzed using conventional flow cytometry and cell sorting (FACS) analysis, as previously described [9]. Neutrophils were stained with anti-mouse FITC-conjugated B220 (clone RA3-6B2; BD Biosciences), NK1.1 (clone PK136; BD Biosciences), CD49b (clone DX-5; BD Biosciences), CD90.2 (clone 53-2.1; BD Biosciences), Ter119 (clone Ter119; BioLegend, San Jose, CA, USA), and APC-Cy7-conjugated CD115 (clone AFS98; BioLegend) antibodies, which were used as lineage markers. Blood cells were stained with APC-conjugated CD45 (clone 30-F11; BioLegend), PerCP-Cy5.5-conjugated CD11b (clone M1/70; BD Biosciences), and PE-conjugated Ly-6G (clone 1A8; BD Biosciences). To stain mature BM neutrophils, anti-mouse FITC-conjugated B220 (clone RA3-6B2; BD Biosciences), NK1.1 (clone PK136; BD Biosciences), CD3e (clone DX-5; BD Biosciences), CD90.2 (clone 53-2.1; BD Biosciences), CD90.2 (clone Ter119; BioLegend), Sca-1 (clone DX-5; BD Biosciences), and Siglec-F (clone DX-5; BD Biosciences) were used as lineage markers. Blood cells were stained with BV711-conjugated c-kit (clone 2B8; BioLegend), BV605-conjugated CD115 (clone CSF-1R; BioLegend), BV421-conjugated CD184 (clone L276F12; BioLegend), PE/Cy7-conjugated Gr-1 (clone RB6-8C5; BioLegend), cy5.5-conjugated CD11b (clone M1/70; BioLegend), PE-conjugated CD182 (clone SA044G4; BioLegend), and AF594-conjugated Ly-6G (clone 1A8; BioLegend) as previously described [30]. For staining with PSGL-1, PE-conjugated CD162 (clone 2PH1; BD Biosciences) was used. To stain aged PB neutrophils, blood cells were stained with PerCP-Cy5.5-conjugated CD11b (clone M1/70; BD Biosciences), PE-conjugated Ly-6G (clone 1A8; BD Biosciences), FITC-conjugated CD62L (clone MEL-14; BioLegend), and BV421-conjugated CD184 (clone L276F12; BioLegend). Aged neutrophils were gated as CXCR4^hi^CD62L^lo^ cells within the neutrophil population, as previously described [31]. Cell counting was performed using a Sony MA900 (Sony Biotechnology Inc., Tokyo, Japan). Data were processed using FlowJo software (BD Biosciences).

### 2.11. In Vitro NET Formation Analysis of Mature BM Neutrophils

BM cells were obtained from the femurs and tibias of 12-week-old defeated and control mice before FeCl_3_ application. After red blood cell sedimentation, mature BM neutrophils (CXCR2^+^Ly-6G^+^) were isolated. An optimal amount of PRP was obtained from group-housed WT male mice. Mature BM neutrophils were stimulated with thrombin-induced activated platelets for 4 h at 37 °C in 5% CO_2_, as previously described [31,32]. Samples were then incubated with primary anti-histone H3 antibody (1:300, citrulline 2,8,17, ab5103; Abcam), Alexa Fluor647–conjugated secondary antibody (1:500, A-21038; Thermo Fisher Scientific), and FITC-conjugated anti-myeloperoxidase antibody (1:50, ab90812; Abcam), as previously described [33].

### 2.12. Statistical Analysis

Data are expressed as mean ± standard error of mean. Normality of the distribution was assessed using the Shapiro–Wilk test or Anderson–Darling test. Equal variances were assessed by Fisher’s test for two groups and Bartlett test or Brown–Forsythe test for more than three groups. Comparisons were performed using the Mann–Whitney test for two groups or the Kruskal–Wallis test for more than three groups if data were not normally distributed. Student’s *t*-test or analysis of variance (ANOVA) followed by the Tukey–Kramer test was used to analyze significant differences between the groups. Significant differences among groups for dependent variables were detected using two-way ANOVA: DNase I treatment and anti-CD11b antibody treatment; otherwise, it is stated in each figure legend. Statistical significance was set at *p* < 0.05. All analyses were performed using GraphPad Prism 8.3.0 Mac (GraphPad Software, LLC, San Diego, CA, USA).

## 3. Results

### 3.1. Fibrin-Rich Clot Formation Was Exaggerated in Defeated Mice

Eight-week-old male WT mice were exposed to RSD and subjected to the tail suspension test and social interaction test (Figure 1A). The total duration of immobility was significantly longer in RSD-exposed mice than in non-exposed mice (Figure 1B). After the social interaction test to confirm depressive-like behaviors, 19 of 31 RSD-exposed mice with an SIR of <1.0 were certified as defeated mice. However, two non-exposed mice with an SIR of <1.0 were excluded in the following study (Figure 1C). Body weight, hemodynamic parameters, and lipid profiles before FeCl_3_ application were comparable between the two groups (Appendix A). In the preliminary experiment, we observed completely occlusive thrombi at 3 h after application (Figure 1D), which is consistent with previously reported findings [27]. Therefore, we examined the volume of thrombi at 3 h after application; however, there was no difference between the two groups (Figure 1E,F). Hematoxylin and eosin staining revealed a remarkable difference in the thrombus components between the two groups (Figure 2A). On examination of the fibrin(ogen)-positive areas, a significant increase was observed in defeated mice (Figure 2B,C). Furthermore, the number of Ly-6G-positive cells in the thrombi was markedly higher in defeated mice than in control mice (Figure 2D,E). In contrast, the percentage of CD42b-positive areas in the thrombi was comparable between the two groups (Figure 2F,G). These findings suggested that segmented accumulation of neutrophils played a crucial role in fibrin-rich clot formation in defeated mice.

### 3.2. DNase I Treatment Inhibited Fibrin-Rich Clot Formation in Defeated Mice

Immunohistochemical analysis was performed to detect NET formation in the thrombi. Ly-6G/Cit-H3 double-positive areas tended to be larger in defeated mice than in control mice (Appendix A). However, quantitative analysis to detect clearly discernable differences between the two groups was technically difficult to perform. Therefore, we examined the effect of NET formation on fibrin-rich clot formation in defeated mice, and DNase I was injected 15 min before FeCl_3_ application (Appendix A) after confirmation of depression-like behavior in mice (Appendix A). The volume of thrombi was not affected by DNase I treatment (Figure 3A,B). The fibrin(ogen)-positive area in defeated mice was significantly reduced by DNase I treatment, similar to that in the control mice (Figure 3C,D). In contrast, the number of Ly-6G-positive cells in the thrombi was higher in defeated mice than in control mice (Figure 3E,F). These findings suggested that NET formation substantially contributed to fibrin-rich clot formation in defeated mice, independent of increased neutrophil accumulation.

### 3.3. Platelet Aggregation and P-Selectin Expression Were Not Affected by RSD

To elucidate the mechanism of augmented NET formation in defeated mice, we focused on the interaction between neutrophils and platelets. The number of peripheral platelets was comparable between the two groups (Figure 4A). Platelet aggregation after stimulation with ADP or collagen did not differ between the two groups (Figure 4B). The expression of P-selectin was markedly increased after thrombin stimulation; however, there was no difference between the two groups (Figure 4C,D). These findings suggested that platelet activation was not likely to be primarily involved in augmented NET formation in defeated mice.

### 3.4. The Number of PB Neutrophils before FeCl_3_ Application Was Not Affected by RSD

The fraction of BM neutrophils and number of PB neutrophils before FeCl_3_ application were comparable between the two groups (Figure 5A,B). However, the number of PB neutrophils was markedly increased after FeCl_3_ application, whereas the BM fraction did not differ between the two groups (Figure 5C,D). These findings suggested that RSD per se did not affect the number of PB neutrophils and that arterial injury via FeCl_3_ application augmented the egress of mature neutrophils from the BM to the PB. To examine the effect of RSD on phenotypic modulation of neutrophils, we sorted the fractions of mature BM and PB neutrophils before FeCl_3_ application in the subsequent experiments.

### 3.5. Activated Platelet-Induced NET Formation Was Augmented in Defeated Mice

To examine in vitro NET formation induced by activated platelets, we first sorted the fraction of PB neutrophils. However, the number of PB neutrophils was too small to analyze NET formation using FACS. Therefore, we sorted mature BM neutrophils (CXCR2^+^Ly-6G^+^), which resemble PB neutrophils [30]. After confirming in vitro NET formation using mature BM neutrophils and thrombin-induced activated platelets in WT mice (Appendix A), we examined in vitro NET formation using mature BM neutrophils of defeated and control mice and thrombin-induced activated platelets from WT mice. NET formation was significantly increased after treatment with thrombin-induced activated platelets in both groups; however, the extent of NET formation was much higher in defeated mice (Figure 6A,B). Immunofluorescence analysis showed that the number of SYTOX Green/Cit-H3 double-positive cells was significantly higher in defeated mice than in control mice (Figure 6C,D). These findings suggested that NET formation elicited by the interaction between neutrophils and platelets was exaggerated in neutrophils of defeated mice.

### 3.6. The CD11b/Gpiα Axis Contributed to Augmented NET Formation in Defeated Mice

Neutrophil–platelet interactions play a crucial role in NET formation, especially between P-selectin and PSGL-1 or between GPIbα and CD11b [32,34,35,36]. Because P-selectin expression after thrombin stimulation was comparable between the two groups (Figure 4C,D), we examined PSGL-1 expression in mature BM neutrophils and PB neutrophils using FACS. PSGL-1 expression levels were significantly higher in PB neutrophils than in mature BM neutrophils. However, PSGL-1 expression levels were lower in defeated mice than in control mice (Figure 7A,B). In contrast, CD11b expression levels in mature BM neutrophils and PB neutrophils were much higher in defeated mice than in control mice, suggesting that CD11b/GPIbα was responsible for augmented NET formation in defeated mice (Figure 7C,D).

### 3.7. Augmented NET Formation in Defeated Mice Was Inhibited by Anti-CD11b Antibody Treatment

To examine the contribution of augmented CD11b expression to NET formation in defeated mice, mature BM neutrophils were treated with anti-CD11b antibody before activated platelet stimulation. NET formation after antibody treatment was markedly reduced in defeated mice, with no statistically significant difference between the two groups (Figure 8A,B). These findings suggested that enhanced CD11b expression in defeated mice played a substantial role in augmented NET formation. To investigate the mechanisms of enhanced CD11b expression in defeated mice, we examined the proportion of peripheral aged neutrophils before FeCl_3_ application. However, there was no difference between the two groups (Appendix A), suggesting that the age-related phenotype was not likely to be associated with augmented CD11b expression in primed neutrophils through RSD exposure.

## 4. Discussion

In this study, we showed for the first time that fibrin(ogen)-positive areas in FeCl_3_-induced thrombi were significantly increased and accompanied by enhanced NET formation in defeated mice. Treatment with DNase I completely diminished the exaggerated fibrin-rich clot formation in defeated mice without affecting the number of Ly-6G-positive cells in the thrombi. Furthermore, in vitro NET formation induced by thrombin-activated platelets was significantly augmented in mature BM neutrophils of defeated mice, which was completely inhibited by anti-CD11b antibody treatment. The expression of CD11b in mature BM and peripheral neutrophils was significantly higher in defeated mice than in control mice. These findings suggest that RSD exaggerates fibrin-rich clot formation after arterial injury by enhancing NET formation via platelet–neutrophil interactions, indicating the possibility that NETs can be a promising therapeutic target in depression-related CVD.

Despite of the equivalent volume of thrombi, fibrin(ogen)-positive areas were markedly increased in defeated mice. Longstaff et al. demonstrated that DNA and histones derived from NETs could modify the structure of fibrin to be more stable and rigid, thereby leading to prolongation of clot lysis times [37]. Sumaya et al. examined the impact of clot properties on adverse outcomes in patients with ACS and showed that clot lysis times independently predicted the 1 year rate of adverse outcomes regardless of randomized antiplatelet treatment [38]. Considering that thrombi obtained from patients with ACS were mainly composed of fibrin [39], NET-mediated clot modification toward fibrin-polymerized network formation might be responsible for the higher risk of ACS in patients with depression than in control subjects.

The interaction between neutrophils and activated platelets has been shown to play a crucial role in promoting thrombosis and inflammation, thereby extending the concept of immunothrombosis [36,40]. Platelet-derived cell adhesion molecules and soluble mediators have been shown to promote NET formation [32]. Walsh et al. were the first to investigate platelet receptor function in patients with depression [41] and showed that the expression levels of glycoprotein (GP) Ib and CD62 (P-selectin) were increased in patients with depression compared to those in control subjects; however, subsequent studies failed to provide conclusive results [42]. We also observed that platelet aggregation on ADP or collagen stimulation and the expression levels of P-selectin on thrombin stimulation were similar between the two groups. Based on the finding that NET formation induced by activated platelets from WT mice was significantly enhanced in defeated mice, we focused on the phenotypic change of neutrophils in terms of their interaction with platelets. The expression of CD11b, but not PSGL-1 (counter receptor for P-selectin), was significantly higher in neutrophils of defeated mice than in those of control mice and treatment with anti-CD11b antibody significantly inhibited augmented NET formation in defeated mice. These findings suggest that conformational changes in integrin αM (CD11b) play a substantial role in enhancing NET formation in defeated mice. Wang et al. have shown that Mac-1 (integrin αMβ2) engagement with platelet GPIbα plays an essential role in thrombus formation after carotid artery injury in Mac-1-deficient and Mac-1-mutant mice [35]. Flick et al. reported that fibrin(ogen) could be a ligand for αMβ2 integrin and serve as a bridging molecule between leukocytes and platelets [43], which is consistent with our findings that fibrin(ogen)-positive areas in thrombi were significantly increased in defeated mice. Taken together, augmented expression of CD11b in neutrophils from defeated mice is likely to not only enhance NET formation but also facilitate interaction with activated platelets via the fibrin(ogen) network.

It has been widely assumed that peripheral neutrophil homeostasis is tightly regulated through egress from the BM and migration back in a circadian manner [11]. Colotta et al. were the first to report that in vitro neutrophil survival was prolonged after stimulation with inflammatory cytokines by the inhibition of programmed cell death [44]. Serum levels of inflammatory cytokines, such as TNF-α and IL-6, were higher in depressed patients than in control subjects [45]. Therefore, the longevity of neutrophils is likely to be extended in defeated mice. Casanova-Acebes et al. were the first to describe a population of CD62L^lo^CXCR4^hi^ neutrophils as “aged neutrophils” [46]. They showed that the number of aged neutrophils was the highest at zeitgeber time 5, which is a period between the active release and clearance of neutrophils. We performed arterial injury and FACS analysis during the same time period. Zhang et al. have shown that neutrophil aging is driven by Toll-like receptor activation along with enhanced Mac-1 activation, leading to augmented NET formation [47]. Furthermore, Uhl et al. demonstrated that lipopolysaccharide-induced integrin affinity was significantly upregulated in aged neutrophils compared to that in non-aged neutrophils, while PSGL-1/CD162 expression was comparable between the two phenotypes [48]. Adrover et al. investigated the molecular mechanisms underlying neutrophils aging and showed that Bmal1 and CXCR2 signaling promoted neutrophil aging, whereas CXCR4 inhibited it [49]. Based on these findings, we examined aged neutrophils in defeated mice. However, we found no difference in the number of aged neutrophils between the two groups before arterial injury despite the increased expression of CD11b in defeated mice. The precise mechanisms of enhanced CD11 expression in neutrophils of defeated mice need to be further investigated in future studies.

Along with the wide use of lipid-lowering therapy during the statin era, plaque characteristics responsible for ACS have shifted from a focus on ruptured plaques to that on eroded plaques with a decline in the incidence of ST-segment elevation myocardial infarction [20,21]. Recently, the molecular characteristics of plaque erosion have been extensively investigated. Plaque erosion is characterized by the presence of a platelet-rich thrombus, in which NET formation plays a crucial role. Quillard et al. were the first to demonstrate that NET formation potentiated endothelial cell apoptosis and detachment, which is strongly implicated in the pathogenesis of superficial erosion [50]. Franck et al. also showed that NET formation contributed to acute thrombotic complications and arterial intimal injury using a flow perturbation model of mouse carotid arteries [51]. Mangold et al. examined the culprit lesion site of coronary thrombi in patients with ACS, in which polymorphonuclear cells were the predominant cell type, which was accompanied by significantly more DNA-histone-positive staining corresponding to NET [52]. NET formation has also been shown to increase thrombus stability through activation of coagulation cascade and crosstalk with activated platelets [53,54,55,56,57,58]. These findings further support our hypothesis that exaggerated NET formation is associated with a higher risk of ACS in patients with depression.

Tsankova et al. were the first to investigate the precise molecular mechanisms of depression using an animal model of a chronic social defeat stress, which mimics many symptoms of depression in humans [18]. They analyzed all mice exposed to stress and examined the effect of chronic social defeat stress on the regulation of brain-derived neurotrophic factor (BDNF) via epigenetic mechanisms. Berton et al. also examined the effect of social defeat stress on gene expression in mesolimbic dopamine pathway using stress-exposed mice independent of stress-induced alteration in depression-related behaviors [25]. Afterward, Golden et al. exhaustively reported a standardized protocol of RSD in mice [23]. They cited the paper by Tsankova et al. [18] and clearly stated that “Historically, a SI ratio equal to 1, in which equal time is spent in the presence versus absence of a social target, has been used as the threshold for dividing categories defeated mice into the susceptible and resilient.” They also showed that 41 (37.6%) out of a total of 109 defeated mice failed to show social avoidance behavior as indicated by their social interaction ratio, which was entirely consistent with our findings that 19 of 31 RSD-exposed mice with an SIR of <1.0 were certified as defeated mice. Because we focused on the effect of depression-related phenotype (susceptible) after RSD on arterial thrombosis by considering its application in a clinical setting, we analyzed only susceptible mice showing depression-like behaviors after RSD exposure.

The gender-specific difference in the effect of psychological stress on the stress-induced alteration in depression-related behaviors and CVD development is critically important. In clinical studies, inflammatory response to stress is enhanced in female subjects, which might partially explain an increased risk of developing stress-related disorders such as depression in females as compared to that in males [59]. However, psychological factors and depression were strongly associated with the incident cardiovascular diseases; however, there was no significant difference between women and men [4,60]. As is the case with clinical data, previous studies showed that female mice were more sensitive to chronic mild stress than male mice as measured by the inflammatory response in central nervous system and behavioral alterations [59]. As far as we know, the RSD stress model applied in this study has not been conducted in female mice; however, RSD-induced alteration in depression-like behaviors and its effect on neutrophil extracellular trap formation via platelet–neutrophil interactions need to be investigated in future studies.

In this study, we focused on the link between psychological stress and innate responses and further examined the interaction between neutrophils and activated platelets in the context of NET formation. We showed that psychological stress could affect clot properties by promoting NET formation via neutrophil and platelet interactions. Our findings provide novel insights into the mechanisms of depression-related CVD development, showing that NET can be a promising therapeutic target in depression-related CVD.

## Figures and Tables

**Figure 1 cells-10-03344-f001:**
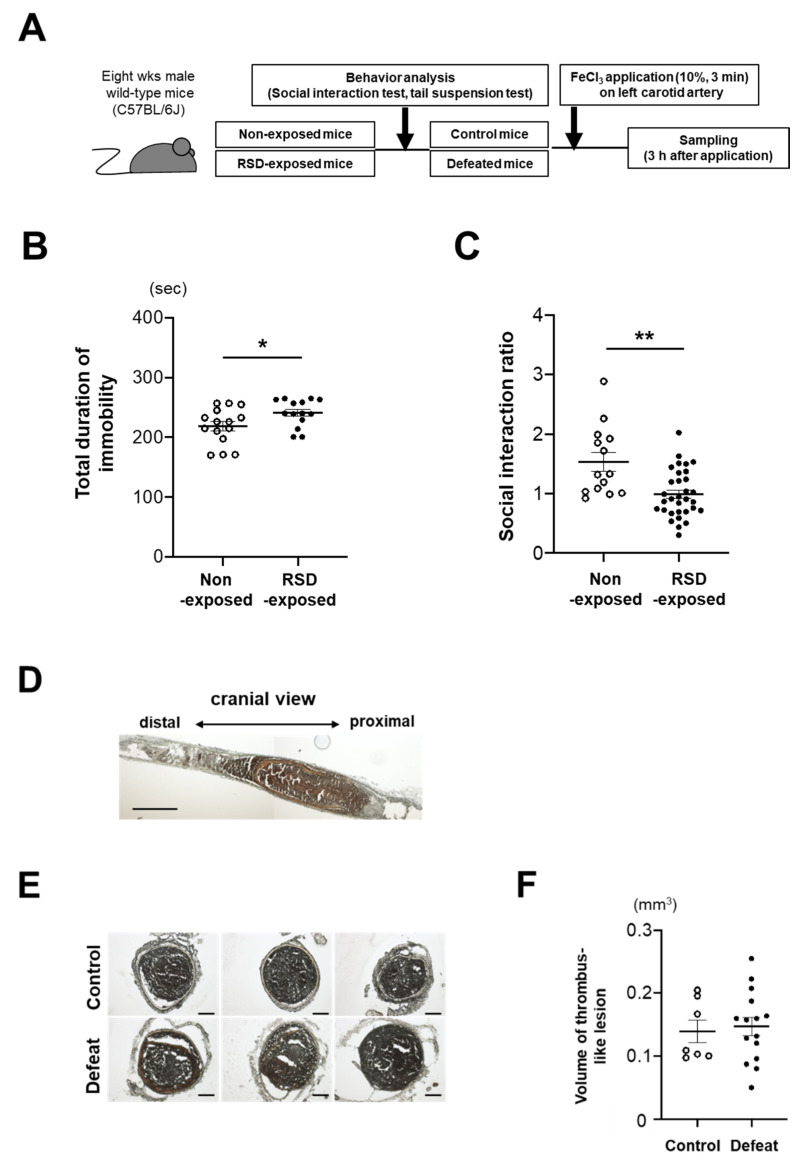
Repeated social defeat does not affect the volume of thrombi after FeCl_3_ application. (**A**) Timeline of experimental protocol. (**B**) Tail suspension test. Values represent mean ± standard error of mean (SEM) for 15 non-exposed mice and 14 RSD-exposed mice before application of FeCl_3_. * *p* < 0.05 vs. non-exposed mice; Student’s *t*-test. (**C**) Social interaction test. Values represent mean ± standard error of mean (SEM) for 14 non-exposed mice and 32 RSD-exposed mice before FeCl_3_ application. ** *p* < 0.01 vs. non-exposed mice; Student’s *t*-test. RSD, repeated social stress. (**D**) Representative photographs of FeCl_3_-induced thrombus in the left carotid artery of wild-type (WT) mice. Scale bar = 500 μm. (**E**) Representative photographs of FeCl_3_-induced thrombus in control and defeated mice. Scale bar = 100 μm. (**F**) Quantitative analysis of the volume of thrombi in control and defeated mice. Values represent mean ± standard error of mean (SEM) for 7 control mice and 15 defeated mice.

**Figure 2 cells-10-03344-f002:**
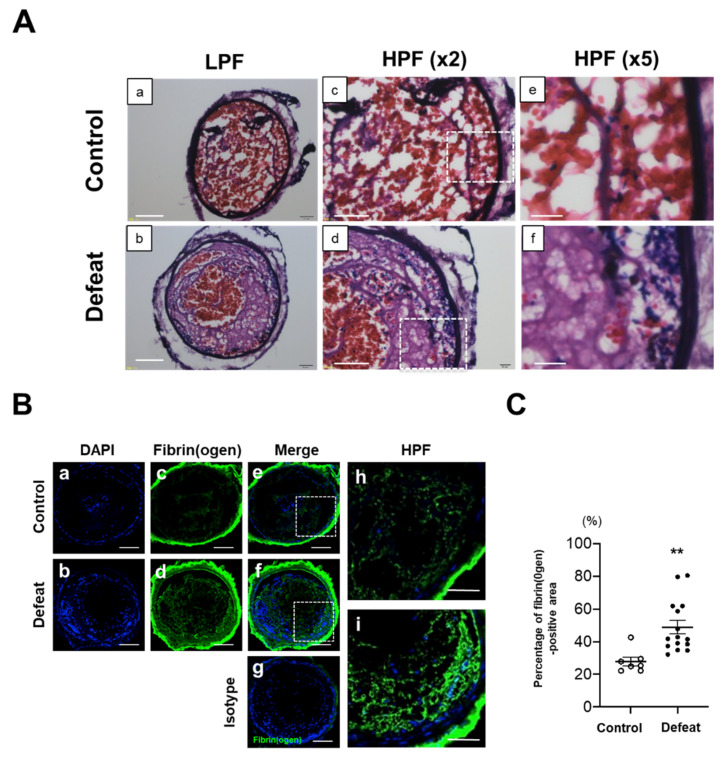
RSD exaggerates the development of fibrin-rich clot formation after FeCl_3_ application. (**A**) Representative photographs of hematoxylin and eosin staining in control and defeated mice. Scale bar = 100 μm (**a**,**b**), 50 μm (**c**,**d**), and 20 μm (**e**,**f**). High magnification images (**e**,**f**) showing the area surrounded by a broken line in panel **c**,**d**, respectively. LPF, low-power field; HPF, high-power field. (**B**,**C**) Representative fluorescent images of fibrin(ogen) and quantitative analysis of the percentage of fibrin(ogen)-positive area. Scale bar = 100 μm (**a**–**g**) and 50 μm (**h**,**i**). High-magnification images (**h**,**i**) showing the area surrounded by a broken line in panel **e**,**f**, respectively. Values represent mean ± standard error of mean (SEM) for 7 control mice and 15 defeated mice. ** *p* < 0.01 vs. control mice; Mann–Whitney test. HPF, high-power field. (**D**,**E**) Representative fluorescent images of Ly-6G and quantitative analysis of the number of Ly-6G-positive cells. Scale bar = 100 μm (**a**–**g**) and 50 μm (**h**,**i**). High-magnification images (**h**,**i**) showing the area surrounded by a broken line in panel (**e,f**), respectively. Arrows indicate Ly-6G-positive cells. Values represent mean ± standard error of mean (SEM) for 7 control mice and 15 defeated mice. * *p* < 0.05 vs. control mice; Mann–Whitney test. HPF, high power field. (**F**,**G**) Representative fluorescent images of CD42b and quantitative analysis of the percentage of CD42b-positive area. Scale bar = 100 μm (**a**–**g**) and 20 μm (**h**,**i**). High-magnification images (**h**,**i**) showing the area surrounded by a broken line in panel **e**,**f**, respectively. Values represent mean ± standard error of mean (SEM) for 7 control mice and 15 defeated mice. HPF, high-power field.

**Figure 3 cells-10-03344-f003:**
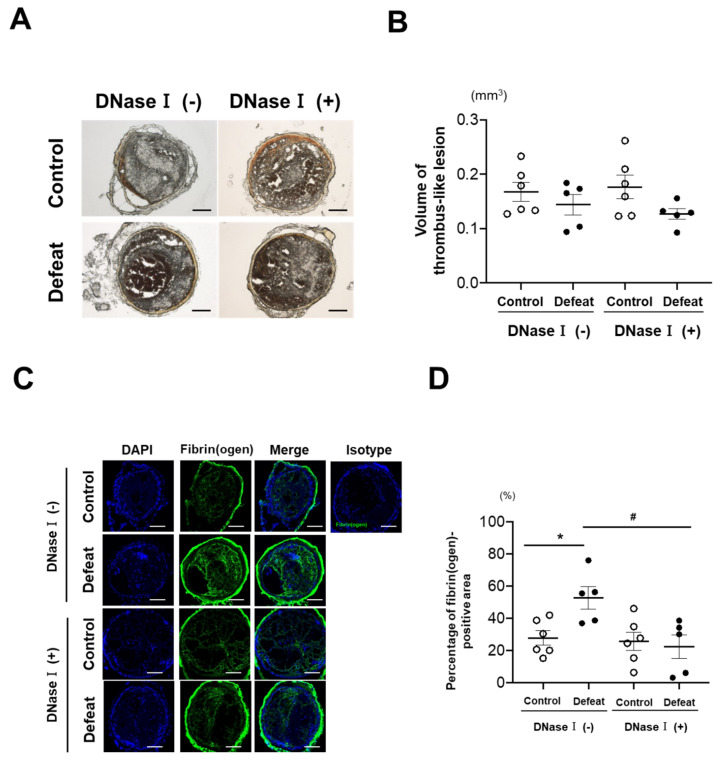
DNase I treatment completely inhibits augmented fibrin-rich clot formation in defeated mice. (**A**,**B**) Representative photographs of FeCl_3_-induced thrombus and quantitative analysis of the volume of thrombi in control and defeated mice with or without DNase I treatment. Scale bar = 100 μm. Values represent mean ± standard error of mean (SEM) for six control and five defeated mice without DNase I treatment and for six control and five defeated mice with DNase I treatment. (**C**,**D**) Representative fluorescent images of fibrin(ogen) and quantitative analysis of the percentage of fibrin(ogen)-positive area. Scale bar = 100 μm. Values represent mean ± standard error of mean (SEM) for six control and five defeated mice without DNase I treatment and for six control and five defeated mice with DNase I treatment. * *p* < 0.05 vs. control mice, ^#^ *p* < 0.05 vs. defeated mice; two-way ANOVA with the Tukey–Kramer post hoc test. (**E**,**F**) Representative fluorescent images of Ly-6G and quantitative analysis of the number of Ly-6G-positive cells. Scale bar = 100 μm. Values represent mean ± standard error of mean (SEM) for six control and five defeated mice without DNase I treatment and for six control and five defeated mice with DNase I treatment. * *p* < 0.05 vs. control mice with DNase I treatment; Kruskal–Wallis test.

**Figure 4 cells-10-03344-f004:**
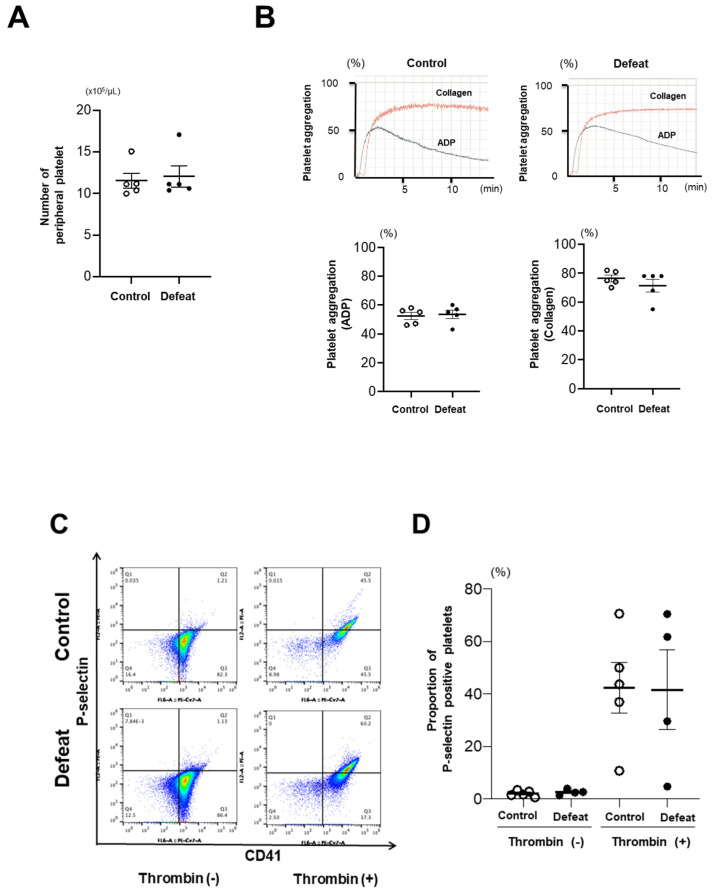
Platelet aggregation and P-selectin expression are comparable between control and defeated mice. (**A**) Number of peripheral platelets before FeCl_3_ application. Values represent mean ± standard error of mean (SEM) for five control and five defeated mice. (**B**) Platelet aggregation on either ADP or collagen activation. The black line represents platelet aggregation on ADP stimulation. The red line represents platelet aggregation on collagen stimulation. Values represent mean ± standard error of mean (SEM) for five control and five defeated mice. ADP, adenosine diphosphate. (**C**,**D**) Flow cytometric analysis of P-selectin expression in platelets from control and defeated mice. Values represent mean ± standard error of mean (SEM) for six control and six defeated mice without thrombin and for six control and five defeated mice with thrombin.

**Figure 5 cells-10-03344-f005:**
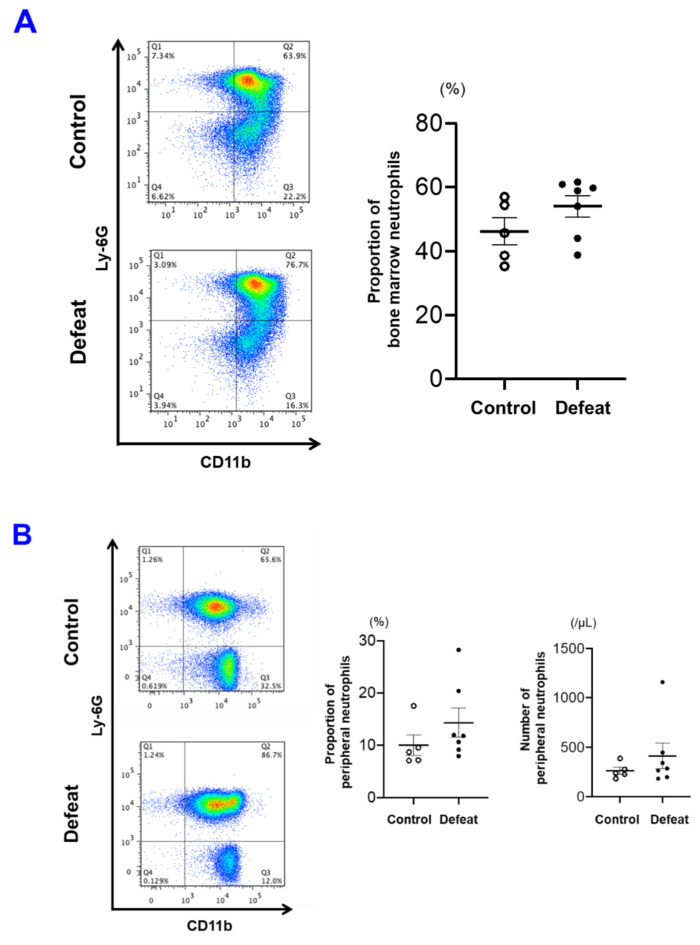
RSD does not affect the number of PB neutrophils before FeCl_3_ application. (**A**,**B**) The number of PB neutrophils was similar between control and defeated mice. Values represent mean ± standard error of mean (SEM) for five control and seven defeated mice for BM neutrophils (**A**) and PB neutrophils (**B**). (**C**,**D**) The number of PB neutrophils after FeCl_3_ application was significantly increased in defeated mice. Values represent mean ± standard error of mean (SEM) for 7 control and 15 defeated mice for BM neutrophils (**C**) and PB neutrophils (**D**). * *p* < 0.05 vs. control mice; Student’s *t*-test. BM, bone marrow; PB, peripheral blood.

**Figure 6 cells-10-03344-f006:**
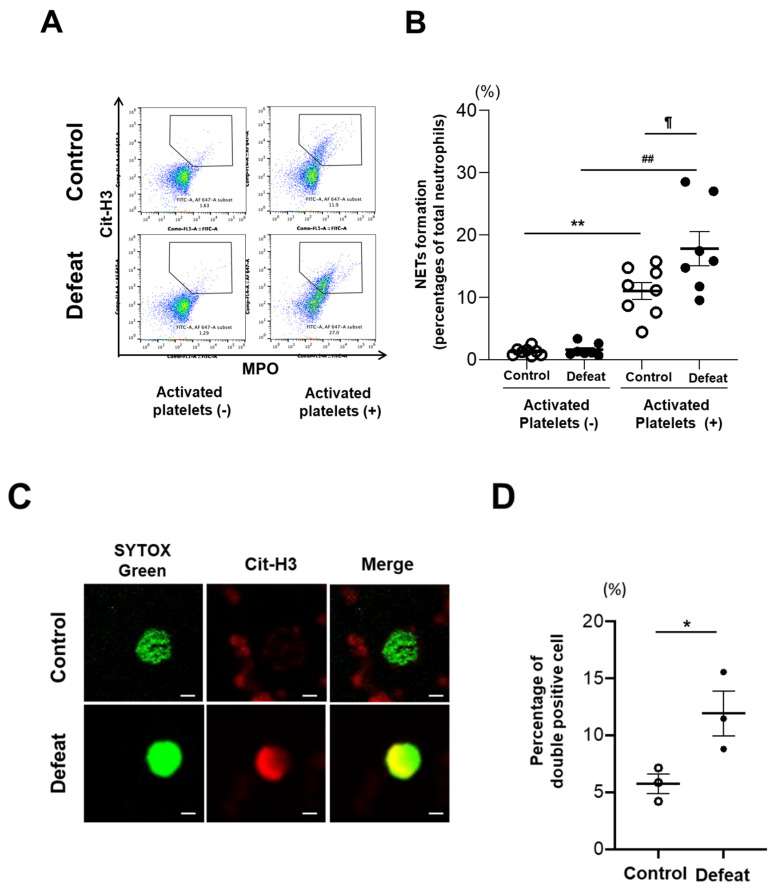
In vitro NET formation induced by activated platelets is exaggerated in defeated mice. (**A**,**B**) Flow cytometric analysis of NET formation induced by activated platelets. Values represent mean ± standard error of mean (SEM) for eight control and seven defeated mice without activated platelets and for eight control and seven defeated mice with activated platelets. ** *p* < 0.01 vs. control mice without activated platelets, ^##^ *p* < 0.01 vs. defeated mice without activated platelets, ^¶^ *p* < 0.05 vs. control mice with activated platelets; two-way ANOVA with the Tukey–Kramer post hoc test. (**C**,**D**) Representative fluorescent images of Cit-H3-positive staining and quantitative analysis of the percentage of Cit-H3-positive cells. Scale bar = 2 μm. Values represent mean ± standard error of mean (SEM) for three control and three defeated mice with activated platelets. * *p* < 0.05 vs. control mice; Student’s *t*-test.

**Figure 7 cells-10-03344-f007:**
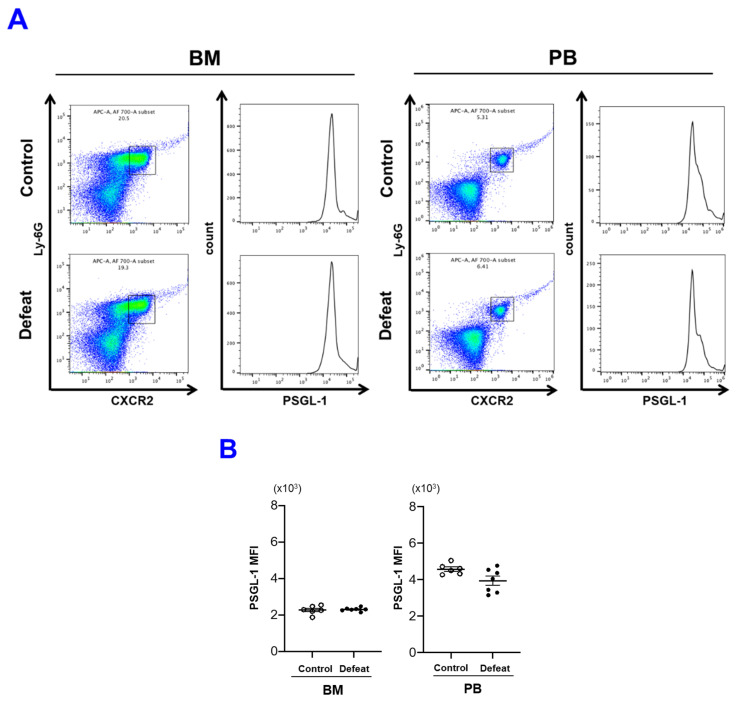
CD11b expression is increased in mature BM neutrophils of defeated mice. (**A**,**B**) Flow cytometric analysis of PSGL-1 expression in mature BM and PB neutrophils from control and defeated mice. Values represent mean ± standard error of mean (SEM) for six control and six defeated mice; Student’s *t*-test. BM, bone marrow; PB, peripheral blood. (**C**,**D**) Flow cytometric analysis of CD11b expression in mature BM and PB neutrophils from control and defeated mice. Values represent mean ± standard error of mean (SEM) for seven control and seven defeated mice. * *p* < 0.05 vs. mature BM neutrophils from control mice; Welch’s test, ^#^ *p* < 0.05 vs. PB neutrophils from control mice; Student’s *t*-test. BM, bone marrow; PB, peripheral blood.

**Figure 8 cells-10-03344-f008:**
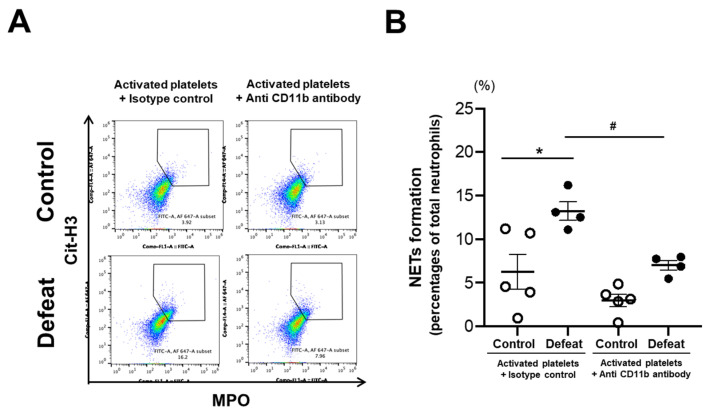
Anti-CD11b antibody treatment markedly inhibits NET formation of mature BM neutrophils in defeated mice. (**A**) Representative flow cytometry data showing Cit-H3- and MPO-positive cells in mature BM neutrophils from control and defeated mice with or without anti-CD11b antibody treatment. (**B**) Quantitative analysis of double-positive cells. Values represent mean ± standard error of mean (SEM) for five control neutrophils and four defeated mice neutrophils with isotype treatment and for five control neutrophils and four defeated mice neutrophils with anti-CD11b antibody treatment. * *p* < 0.05 vs. neutrophils from control mice with isotype treatment, ^#^ *p* < 0.05 vs. neutrophils from defeated mice with isotype treatment; two-way ANOVA with the Tukey–Kramer post hoc test. Cit-H3, citrullinated histone H3; MPO, myeloperoxidase.

## Data Availability

Not applicable.

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
