# Peer review of "Repeated Social Defeat Exaggerates Fibrin-Rich Clot Formation by Enhancing Neutrophil Extracellular Trap Formation via Platelet–Neutrophil Interactions"

_cells, 2021, doi:10.3390/cells10123344_

Round 1

Reviewer 1 Report

The manuscript from Sugimoto et al. found that the FeCl3-induced thrombus is rich in fibrin(ogen) in the defeated mice, whereas the thrombi volume is the same between control and defeated groups. In addition, they show that the fibrin(ogen)-rich thrombi in defeated mice should be related to the formation of neutrophil extracellular trap, independent of neutrophil accumulation. The manuscript is well-written, the experimental design is robust and support some of the authors' conclusions, but the authors should consider the following issues:

Major:

  • The manuscript would benefit from a full description of the behavioral tests and apparatus used in the work.
  • In the SIR methods, the authors describe that “RSD-exposed mice exhibiting an SIR of < 1.0 were considered as defeated mice with depression-like behaviors, while non-exposed mice exhibiting an SIR of > 1.0 were considered as control mice” (lines 100-104). For me, the way this is described suggests that mice not exhibiting the values above in the respective group were excluded from the analysis. Were the animals falling out of the classification excluded from the study? If it is the case, the authors should present the rationale for excluding the animals from the study. This is particularly important since it seems that a large number of animals is excluded in each experimental set and some conclusions were drawn based on the depressive-like behaviors.
  • In the abstract and at the beginning of the discussion (lines 425-427) the authors suggest that NETs can be a promising therapeutic target in depression-related CVD. However, data of mice exposed to social defeat that did not show behavioral alterations (i.e. resilient mice) are not shown. Do the authors have the data for those animals? If it is not the case, the discussion and conclusion in the paper should be focused on the effects of stress exposure rather than focusing on depressive-like behaviors.
  • The authors did not use females in their study, although the use of both sexes is an important practice when studying biological variables and modeling diseases that affect both men and women in clinical practice. I understand that this could be methodologically unfeasible sometimes, but I suggest that authors should discuss this limitation of their study in the discussion section, and possible implications for their findings.

Minor:  

  • Ethical statements could be placed earlier in the text (in addition to the “Institutional Review Board Statement” topic).
  • Were control mice housed with other WT mice or with a CD1 mouse? It should be clearly described in the text.
  • Line 166-168: was the platelet-rich plasma discarded or used in the analysis of P-selectin expression? Please clarify.
  • Figure 3: the order of the panels should be revised (panels E and F are presented before panels C and D).

Author Response

Thank you very much for your careful reading of our manuscript and helpful comments. Please find the attached file below.

Reviewer 2 Report

  1. The authors prepared the manuscript with the focus on the interaction between NET and clot formation pathophysiology. They describe repeated social defeat model , problematic behavior analysis, FeCl3-induced clot formation model, thrombi measurement and histological analysis, immunohistochemical analysis, compex examination of blood count and platelet function including the use of point-of-care tests, such as flow cytometry. Moreover, they added in vitro NET formation analysis of mature BM neutrophils and globally, they described the steps of the study in detail and in a logical manner. Moreover, they used modern methods enabling analysis at the molecular level.

    For all these reasons, I recommend the publication of the article in the current fom.

Author Response

Thank you for your careful reading and kind comments. All authors really appreciate your encouraging evaluation.